# Relationship of preoperative oral hypofunction with prognostic nutritional index in gastric cancer: A case-control retrospective study

**Atsushi Abe**[1], **Atsushi Nakayama**[2]*, **Yuya Otsuka**[1], **Kanae Shibata**[1], **Yoshihito Matsui**[1], **Yu Ito**[1], **Hiroki Hayashi**[1], **Moeko Momokita**[1], **Shinichi Taniguchi**[1]

1 Department of Oral and Maxillofacial Surgery, Nagoya Ekisaikai Hospital, Nagoya, Aichi Prefecture, Japan,
2 The First Department of Oral and Maxillofacial Surgery, School of Dentistry, Aichi-Gakuin University, Nagoya, Aichi Prefecture, Japan

* nkym8020@dpc.agu.ac.jp

**Data Availability Statement:** Informed consent was obtained from the patients. However, the consent does not contain provisions for the release of raw data. Therefore, the data are not freely

## Abstract

Preoperative nutritional status is an important prognostic factor in gastric cancer patients. This study will evaluate whether preoperative oral dysfunction is associated with prognostic nutrition index (PNI). This case-control study analyzed 95 patients who underwent oral function management. We assessed the following parameters: body mass index, stage of gastric cancer, C-reactive protein, total lymphocyte count, albumin, and prognostic nutritional index. The patients were divided into groups with prognostic nutritional indexes <45 and >45. Logistic regression analysis was used to assess the association between the measurements of oral function and the prognostic nutritional index. Univariate analysis of factors associated with decreased oral function and prognostic nutritional index showed significant differences between the two groups in C-reactive protein, neutrophils, and tongue pressure (p<0.01). However, oral hygiene, oral dryness, occlusal force, tongue–lip motor function, masticatory function, and swallowing function were not significantly different. Multivariate analysis showed that C-reactive protein (odds ratio: 0.12, 95% confidence interval: 0.30–0.45, p<0.01) and tongue pressure (odds ratio: 3.62, 95% confidence interval: 1.04–12.60, p<0.05) were independent risk factors for oral hypofunction. Oral function decreased in perioperative patients with gastric cancer, and decreased tongue pressure is associated with a decreased prognostic nutritional index.

## Introduction

Nutritional status and systemic inflammatory response in gastric cancer patients influence disease recovery [1]. The presence of low nutritional status and systemic inflammatory response also increases the incidence of postoperative complications and hospital mortality [2,3]. Systemic inflammatory responses and malnutrition are caused by inadequate dietary intake, reduced activity, oxidative stress, increased inflammatory cytokines, and hormonal imbalances

available in manuscripts, information files, or public repositories. Datasets from this study may be requested from the Clinical Trial Management Office, Nagoya Ekisaikai Hospital, 4-66 Shonen-cho, Nakagawa-ku, Nagoya City, 454-8502, Japan, E-mail: akane1130.com@gmail.com. All requests must be accompanied by a plan detailing the analyses to be performed on the data and must be approved by the hospital's steering committee.

**Funding:** The authors received no specific funding for this work.

**Competing interests:** The authors have declared that no competing interests exist.

[4,5]. At the same time, the progression of periodontal disease has systemic effects such as TGF-β1, VEGF, and endothelial cell-specific markers. These biomarkers are expected to detect the progression of periodontal disease at an early stage and help in planning treatment and predicting prognosis. On the other hand, biomarkers require reliability and simplicity in diagnosis [6–8]. The indicators calculated from the preoperative blood sampling data are used to assess the nutritional status objectively [9]. Currently, the prognostic nutritional index (PNI) is associated with prognosis in several cancers [3,10–13]. The PNI is calculated from serum albumin (ALB) levels and total lymphocyte counts in peripheral blood and has been proposed for assessing the perioperative immune nutritional status and surgical risk of patients scheduled for gastrointestinal surgery. Low PNI may lead to increased postoperative complications and worsened prognosis [14].

Oral function plays a major role in the feeding and swallowing process; moreover, oral dysfunction worsens the nutritional status [15–17] and is significantly associated with sarcopenia, low nutrition, and hypotrophy. Decreased oral function decreases the quantity and quality of food intake, leading to muscle hypotrophy and weight loss, which lead to a negative cascade of events that promote further progression of hypotrophy and sarcopenia. The individual functions of the parts of the oral cavity work complementarily; therefore, the decline in oral function is not detected early and progresses gradually. Thus, oral hypofunction (OHF) assessment has been proposed to quantitatively evaluate functional decline, leading to early detection of oral hypofunction [18,19]. The diagnosis of oral hypofunction is based on seven tests, and a diagnosis is made when the results of three or more of the seven tests are below normal values. The tests include the assessment of oral moisture, bite force, tongue–lip motor function, tongue pressure, masticatory function, and swallowing function. Previous studies have associated poor oral hypofunction with malnutrition, sarcopenia, and frailty [20–23]. However, the studies were conducted on institutionalized elderly patients, and only a few studies have correlated preoperative nutritional status and oral function decline in patients with gastric cancer. This study aimed to examine the association of preoperative oral dysfunction in gastric cancer patients with PNI.

## Materials and methods

### Participants

A case-control retrospective study of 112 patients diagnosed with primary gastric cancer who underwent oral functional management at the Nagoya Ekisaikai Hospital from Jan 1, 2014, to Dec 31, 2021, was conducted. The inclusion criteria were as follows: (1) no history of other cancers, (2) histopathologically diagnosed gastric cancer, (3) no history of radiation therapy and/or chemotherapy, and (4) available preoperative blood test results. The exclusion criteria included the following: (1) history of any preoperative therapy, (2) concurrent infection, (3) steroid use, (4) autoimmune disease, (5) metabolic diseases such as diabetes, (6) recurrence, (7) presence of gastrointestinal stromal tumors, (8) Sjogren's syndrome, and (9) no preoperative blood cell count records, (17 patients). Thus, 95 patients (38 male and 57 female individuals) were enrolled in the study. Patients' clinical data were recorded using a chart review. No *a priori* sample size calculation was required, as this was a case-control retrospective study. This research was independently reviewed and approved by the Ethics Committee of the Nagoya Ekisaikai Hospital (approval number: 2021–048). The study was performed in accordance with the Strengthening the Reporting of Observational Studies in Epidemiology (STROBE) guidelines and the principles of the Declaration of Helsinki. The STROBE checklist is shown in Fig 1. The experiments were undertaken with each participant's understanding and written consent and according to the principles mentioned above.

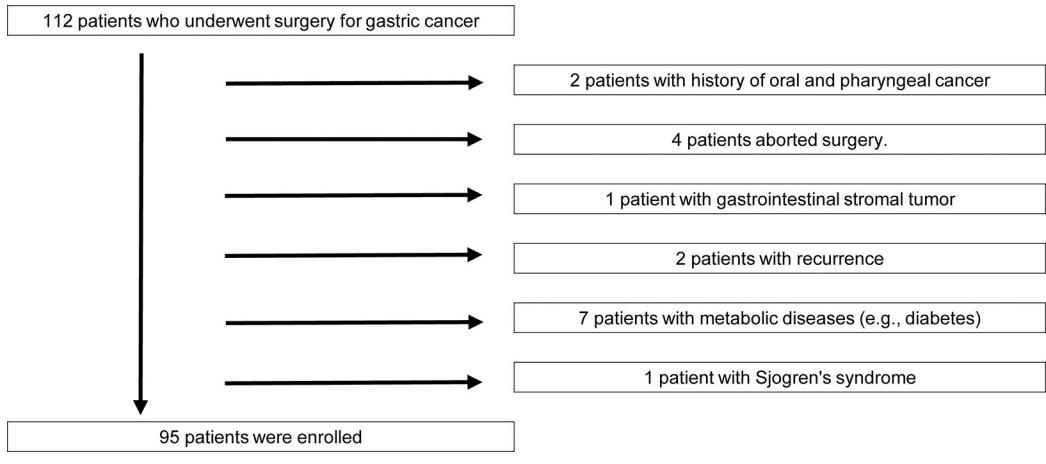

**Fig 1. Flow diagram of exam selection.**

## Evaluation and general status of patients with gastric cancer

Patients with gastric cancer were diagnosed by endoscopy, computed tomography, and magnetic resonance imaging in the gastrointestinal surgery department of Nagoya Ekisaikai Hospital. Nutritional status was determined by calculating the body mass index (BMI), and the ALB levels, C-reactive protein (CRP) levels, and total lymphocyte counts were measured by preoperative blood tests. All blood specimens were collected at most 1 month prior to the preoperative examination for gastric cancer surgery. The patients' height and weight were measured during their first visit to our clinic. The PNI indicates the nutritional status and systemic inflammatory response. In this study, Onodera's method, which involves a simple formula, was used to calculate the PNI. The PNI is a simple scoring system calculated based on the serum ALB level and peripheral blood lymphocyte count [24] by the following formula.

$$PNI = [10 \times serum\ albumin\ level\ (g/dL)] + 0.005 \times total\ peripheral\ lymphocyte\ count(per\ mm^3)].$$

The PNI is typically quantitative and estimates postoperative complications in surgical patients, with a PNI >45 indicating surgery, 45–40 indicating caution during surgery, and <40 indicating a contraindication for surgery [25,26].

## Evaluation of oral condition

A preoperative oral examination was performed to reduce the risk of pneumonia, wound infection, intraoperative tooth loss, and various complications related to the oral region. These parameters were selected for evaluation, as oral bacteria and oral functions, such as chewing, swallowing, and tongue movement functions, are strongly related to these complications.

Two calibrated dentists with more than 7 years of clinical experience performed oral examinations. Based on the diagnostic criteria of the Japanese Society of Geriatric Dentistry, seven sub-symptoms of OHF were measured and evaluated at the dental outpatient clinic on the day or 2 days before the surgery. The seven items include (1) oral hygiene, (2) oral dryness, (3) occlusal force, (4) tongue–lip motor function, (5) tongue pressure, (6) masticatory function, and (7) swallowing function.

To assess oral hygiene, the degree of tongue coating was visually evaluated by the tongue coating index (TCI) [27]. The tongue surface was divided into nine sections, and the degree of tongue coating in each section was rated on a scale of 0–2 points, with 0 indicating the least amount of

tongue coating. The total score for all the sections was calculated as the TCI. A total score of $\geq 9$ points indicated poor oral hygiene [19]. The percentage TCI was calculated as the sum of the scores divided by the maximum score multiplied by 100%. Oral dryness was measured by the wetness of the oral mucosa using an oral moisture meter (Mucus®, Life Co., Ltd., Saitama, Japan) [28,29]. A moisture meter reading of <27 was considered oral dryness. A tongue pressure probe attached to a digital tongue pressure measuring device (JMS Tongue Depressor®, G.C., Tokyo, Japan, JMS) was pressed against the tongue and palate with maximum force. A mean value of <30 kPa was defined as low tongue pressure [17,30]. The participants were instructed to clench the pressure-sensitive film with their teeth (Dental Prescale II®, G.C. Co., Ltd., Tokyo, Japan) for 3 s, and the color change on the pressure-sensitive film was analyzed and converted into occlusal force using the supplied software. A bite force measurement of <200 N was considered a decreased bite force [19]. The tongue and lip movement function was measured by instructing the participants to pronounce the sounds /pa/, /ta/, and /ka/ as many times as possible in 5 s, and the number of consecutive pronunciations of /pa/, /ta/, and /ka/ was measured with an oral function measurement device (Kenkuchi-kun Handy, Takei Kikai Kogyo, Niigata, Japan). For the masticatory function, the participants chewed 2 g of gummy jelly (Glucoram, G.C., Tokyo, Japan) for 20 s, and the gummy jelly was gargled with 10 mL of water. The chewed gummies and water were drained through a filtration mesh, and the solution that passed through the mesh was sampled with an applicator to measure the amount of glucose eluted with a chewing ability testing device (Glucosensor GS-II, G.C., Tokyo, Japan). A glucose density of <100 mg/dL was considered masticatory dysfunction [31]. Swallowing function was evaluated using the Swallowing Screening Questionnaire (a 10-item Eating Assessment Tool [EAT-10], 40-point scale); a total score of $\geq 3$ points was considered as a decline in swallowing function [19].

## Data analysis

Univariate analysis was performed to examine the association between the PNI and oral function. Next, multivariate analysis was performed using selected prognosis-related factors. Patient characteristics and their relationship to the PNI were analyzed using a t-test or ANOVA. A PNI of $\geq 40$ was used as the objective variable; five patient characteristic variables (age, sex, smoking, alcohol consumption, the number of current teeth, Eichner classification) and the seven variables that constitute the oral function test were used as explanatory variables in a logistic regression analysis using the stepwise variable reduction method (Wald; probability of elimination: 0.10). The Wald test was applied to test the regression coefficients. The significance level for all statistical analyses was set at 5% on both sides, and the analyses were conducted by forcibly selecting explanatory variables. All statistical analyses were performed using EZR (Saitama Medical Center, Jichi Medical University, Japan) [32].

## Results

### Clinical characteristics of the patients

Among the 112 patients with gastric cancer who consented to participate, 95 (38 male and 57 female individuals) were included in the present analysis. The characteristics of the patients are shown in Table 1. The age of the participants ranged from 37 to 89 years with a mean age ±standard deviation (SD) of 67.2±13.2 years; their BMI ranged from 14.8 to 43.2 kg/m$^2$ with a mean±SD of 22.4±4.1 kg/m$^2$. The number of patients with stage I, II, III, and IV gastric cancer was 33 (34.7%), 25 (26.3%), 21 (22.1%), and 16 (16.8%), respectively. No significant between-group differences were observed in sex, age, stage of gastric cancer, or BMI. Patients with PNI <40 (24 patients, 25.3%) were classified into the low PNI group, and those with PNI >40 (71 patients, 74.7%) were classified into the high PNI group [33].

**Table 1. Patient background.**

| Characteristics | Group | Overall |
|---|---|---|
| n | | 95 |
| Age (years) | | 67.2±13.2 |
| Sex (%) | Male | 38 (40.0) |
| | Female | 57 (60.0) |
| Smoking (%) | Yes | 53 (56.0) |
| | No | 42 (44.0) |
| Alcohol consumption (%) | Yes | 51 (53.3) |
| | No | 44 (46.7) |
| BMI (kg/m$^2$) | | 22.4±4.1 |
| Albumin | | 3.8±0.7 |
| CRP | | 1.0±1.9 |
| PNI | | 44.9±8.3 |
| WBC | | 6.2±1.9 |
| Neutrophil | | 65.2± 9.8 |
| Total lymphocytes | | 1490.9±566.2 |
| T (%) | 1 | 31 (32.6) |
| | 2 | 23 (24.3) |
| | 3 | 27 (28.0) |
| | 4 | 14 (14.7) |
| N (%) | 0 | 73 (76.9) |
| | 1 | 14 (14.7) |
| | 2 | 6 (6.3) |
| | 3 | 2 (2.1) |
| Stage (%) | 1 | 33 (34.7) |
| | 2 | 25 (26.3) |
| | 3 | 21 (22.1) |
| | 4 | 16 (16.8) |
| Residual teeth | | 18.8±9.0 |
| Eichner classification (%) | A | 48 (50.5) |
| | B | 28 (29.5) |
| | C | 19 (20.0) |
| Oral hygiene | | 24.6±21.0 |
| Oral dryness | | 25.9±4.2 |
| Occlusal force | | 396.8±314.3 N |
| Tongue and lip movement function /pa/ta/ka/ | Yes | 29 (30.5) |
| | No | 66 (69.5) |
| Tongue pressure | | 31.9±11.0 kPa |
| Masticatory function | | 143.5±75.5mg/d L |
| Swallowing function | No problem | 82 (86.7) |
| | Problem | 13 (13.7) |
| Oral hypofunction (%) | Yes | 24 (25.3) |
| | No | 71 (74.7) |

BMI: Body mass index; CRP: C-reactive protein; PNI: Prognostic nutritional index; WBC: White blood cell.

## Oral function

The percentage TCI ranged from 0 to 100% (24.6±21.0%), oral dryness ranged from 8.2 to 32.7 (25.9± 4.2), tongue pressure ranged from 6 to 62.1 kPa (31.9±11.0 kPa), and the occlusal force ranged from 44.9 to 1585.6 N (396.8±314.3 N). The mean value of tongue and lip movement function was less than six times in 66 patients (69.5%). Decreased masticatory function ranged from 24.0 to 494.0 mg/dL (143.5±75.5 mg/dL). The total score for decreased swallowing function was 0–14 points, with 82 patients (86.3%) having a score of ≤3 points, and 13 patients (13.7%) >3 points. In addition, 24 patients (25.3%) had dysfunction in more than three items of oral function and were considered to have OHF. We performed a univariate analysis of factors related to PNI for oral dysfunction. The following variables were analyzed: age, sex, stage of gastric cancer, CRP, oral hygiene, oral dryness, tongue pressure, occlusal force, tongue and lip movement function, masticatory function, and swallowing function. High oral contamination, such as a TCI of >50%, was observed in four (4.2%) and 12 (12.6%) patients in the low and high PNI groups, respectively. Dry mouth was observed in 12 (12.6%) and 27 (28.4%) patients in the low and high PNI groups, respectively. Low tongue pressure was observed in 15 (15.8%) and 24 (25.3%) patients in the low and high PNI groups, respectively. The decreased occlusal force was observed in 11 (11.6%) and 19 (20.0%) patients in the low and high PNI groups, respectively. Decreased tongue and lip motor function were observed in 18 (18.9%) and 48 (50.5%) patients in the low and high PNI groups, respectively. Masticatory dysfunction was observed in 11 (11.6%) and 19 (20.0%) patients in the low and high PNI groups, respectively. Swallowing dysfunction was observed in five (5.3%) and eight (8.4%) patients in the low and high PNI groups (Table 2). Univariate analysis showed significant differences between the two groups' CRP levels, neutrophil count, and tongue pressure ($p < 0.01$; Table 3). Multivariate analysis showed that the CRP levels (odds ratio: 0.12, 95% confidence interval [CI]: 0.30–0.45, $p < 0.01$) and tongue pressure (odds ratio: 3.62, 95% CI: 1.04–12.6, $p < 0.05$) were independent risk factors for PNI (Table 3).

## Discussion

In this study, decreased tongue pressure was associated with decreased PNI, suggesting that poor oral function may be a prognostic factor in gastric cancer. This case-control retrospective study investigated the relationship between PNI and OHF in patients with gastric cancer preoperatively. A low PNI is an independent risk factor for poor prognosis in such patients. Consequently, patients with a low preoperative PNI must be monitored closely after surgery to avoid postoperative complications [3]. Previously, we reported that lower occlusal support levels are associated with a lower PNI [34,35]. However, the assessment of bite support range only evaluates a part of the complex oral function, such as swallowing and mastication. Regarding mastication, the risk of low nutrition and malnutrition can be prevented by devising different food forms and providing appropriate meals for patients. Therefore, in this study, we evaluated the general oral function. The results showed that high CRP and low tongue pressure were associated with a decreased PNI. Decreased tongue pressure was considered to have a possible influence on low nutrition and hidden sarcopenia. The preoperative state of nutrition in patients with gastric cancer is associated with impaired immune system function, delayed wound healing, development of complications, and increased mortality [1–3]. In addition, poor oral function can lead to loss of appetite and dysphagia, making the patient more prone to poor nutrition. A variety of factors, such as OHF can cause preoperative nutritional insufficiency. Although studies have examined the relationship of nutritional status with the number of teeth loss and masticatory function [36,37], to our knowledge, no study has related the preoperative nutritional status in gastrointestinal cancer surgery to comprehensive oral functions

**Table 2. Univariate analysis.**

| Factor | Group | PNI | | p.value |
|---|---|---|---|---|
| | | Low(40<) | High(40≧) | |
| Gender | Male | 14 (14.7) | 24 (25.3) | 0.103 |
| | Female | 10 (10.5) | 47 (49.5) | |
| BMI | Low (<25) | 16(16.8) | 57(60.0) | 0.345 |
| | High (25≧) | 8(8.4) | 14(14.7) | |
| Smoking | Yes | 14 (14.7) | 39 (41.1) | 1 |
| | No | 10 (10.5) | 32 (33.7) | |
| Alcohol | Yes | 10 (10.5) | 41 (43.2)) | 0.296 |
| | No | 14 (14.7) | 30 (31.6) | |
| Stage | 2≦ | 10 (10.5) | 48 (50.5) | 0.0592 |
| | 3≧ | 14 (14.7) | 23 (24.2) | |
| WBC | Low (<) | 18 (18.9) | 58 (61.1) | 0.51 |
| | High (≧) | 6 (6.3) | 13 (13.7) | |
| Neutrophil | Low (<) | 18 (18.9) | 66 (69.5) | 0.0406 |
| | High (≧) | 6 (6.3) | 5 (5.3) | |
| Total lymphocytes | Low (<) | 21 (22.1) | 51 (53.7) | 0.133 |
| | High (≧) | 3 (3.2) | 20 (21.1) | |
| CRP | Low (<5) | 5 (5.3) | 51 (53.7) | 0.000273 |
| | High (5≧) | 19 (20.0) | 20 (21.1) | |
| Residual teeth | 20< | 14 (14.7) | 28 (29.5) | 0.188 |
| | 20≧ | 10 (10.5) | 43 (45.3) | |
| Eichner classfication | A | 8 (8.4) | 40 (42.1) | 0.137 |
| | B | 9 (9.5) | 19 (20.0) | |
| | C | 8 (8.4) | 11 (11.6) | |
| Oral hygiene | Good | 20 (21.1) | 59 (62.1) | 1 |
| | Poor | 4 (4.2) | 12 (12.6) | |
| Oral dryness | Yes | 12 (12.6) | 27 (28.4) | 0.578 |
| | No | 17 (17.9) | 39 (41.1) | |
| Occlusal force | Low (<200) | 11 (11.6) | 19 (20.0) | 0.153 |
| | High (200≧) | 13 (13.7) | 52 (54.7) | |
| tongue–lip motor function ODK /pa/ta/ka/ | Low (<3) | 18 (18.9) | 48 (50.5) | 0.777 |
| | High (3≧) | 6 (6.3) | 23 (24.2) | |
| Tongue pressure | Low (<30) | 15 (15.8) | 24 (25.3) | 0.0328 |
| | High (30≧) | 9 (9.5) | 47 (49.5) | |
| Masticatory function | Low (<100mg/dL) | 11 (11.6) | 19 (20.0) | 0.153 |
| | High (100mg/dL≧) | 13 (13.7) | 52 (54.7) | |
| Swallowing function | No problem | 19 (20.0) | 63 (66.3) | 0.262 |
| | Problem | 5 (5.3) | 8 (8.4) | |
| Oral Hypofunction | Yes | 19 (20.0) | 63 (66.3) | 0.262 |
| | No | 5 (5.3) | 8 (8.4) | |

BMI: Body mass index; CRP: C-reactive protein; ODK: Oral diadochokinesis; PNI: Prognostic nutritional index; WBC: White blood cell.

such as swallowing, mastication, and occlusion. The Japanese Society of Geriatric Dentistry has proposed a diagnostic criterion consisting of seven items as an index to evaluate the decline in oral function comprehensively [19]. If three of the seven items exceed the standard value, the patient is diagnosed with decreased oral function [38]. In this study, 30 patients

**Table 3. Multivariate logistic regression: Logistic regression analysis with oral hypofunction as the objective variable.**

| Factor | Odds ratio | 95% CI | p-value |
|---|---|---|---|
| BMI | 0.32 | 0.06–1.64 | 0.17 |
| Smoking | 1.23 | 0.32–4.72 | 0.76 |
| Alcohol consumption | 0.49 | 0.13–1.85 | 0.29 |
| Stage | 0.26 | 0.06–1.12 | 0.071 |
| CRP | 0.12 | 0.03–0.45 | 0.0015 |
| Tongue pressure | 3.62 | 1.04–12.6 | 0.044 |

BMI: Body mass index; CRP: C-reactive protein; CI: Confidence interval.

(40.0%) were diagnosed with decreased oral function, indicating that many gastric cancer patients had problems before surgery.

Oral functions often require compensatory movements that complement each other, and it is necessary to identify the most influential factors related to oral function. Oral uncleanliness can lead to an abnormal increase in microorganisms in the oral cavity, causing pneumonia, postoperative infections, and oral infections. A higher TCI indicates a higher number of microorganisms on the tongue. Poor oral hygiene of the tongue diminishes the sense of taste and affects food quality. Specifically, poor oral hygiene may affect taste intensity, but it is unlikely to be directly related to nutritional status. No significant difference was found between the two groups in this study, suggesting that the effect of poor oral hygiene on PNI is small. Oral dryness alters the oral environment's homeostasis and induces various disorders. There were no significant differences in oral dryness and PNI between the groups in this study. Occlusal support, residual tooth count, and muscle strength affect occlusal strength [38–40]. The occlusal force and masticatory capability decrease as the number of teeth decreases. According to many reports, oral function can be maintained if the remaining teeth are ≥20 and the occlusal force is ≥200 N [41]. These reports suggest that a decrease in the number of teeth in patients with gastric cancer causes an alteration in diet that leads to low nutrition due to impaired gastrointestinal absorption caused by gastric cancer. The present study found no association between bite force and PNI.

Decreased tongue and lip motor function may affect eating behavior, nutrition, life function, and quality of life. The value of oral diadochokinesis varies depending on the condition of the oral cavity, such as dentures, individual differences in aging changes, underlying diseases, mental status, living environment, nutritional status, and the presence of oral function training [42]. The present study found no significant association between oral diadochokinesis and PNI. The reason may be that the frailty in patients for whom surgery had been indicated had not progressed. Decreased tongue pressure due to chronic dysfunction of the lingual muscle groups decreases the pressure created between the tongue, palate, and food. The tongue performs complex mastication, swallowing, and phonation movements in coordination with the lips, mandible, pharynx, and larynx. These are important functions for maintaining life and quality of life but are difficult to quantify because of the complexity of the movements. The tongue pressure test is one of the few methods to quantify tongue strength. Previous studies have shown that tongue pressure positively correlates with the food intake rate and that individuals with a maximized tongue pressure of ≥30 kPa can consume regular food, while individuals with a low tongue pressure have significantly lower food intake rates and eating patterns [17,30,43]. Our study showed that tongue pressure was an independent risk factor for PNI. When there is poor occlusion, it is assumed that the patient eats a soft diet and swallows by crushing the food with the tongue, jaw crest, and palate. Therefore, with appropriate tongue

pressure, the compensatory mechanisms of oral function would work effectively, and the PNI would not decrease. In contrast, with low tongue pressure, the tongue cannot press against the jaw crest and palate, and compensatory movements are not possible, leading to a low PNI. Therefore, low tongue pressure is a risk factor for dysphagia and low nutrition and may also affect eating patterns, assessment, interventions (such as appropriate and adequate exercise therapy), and therapeutic intervention (such as improvement of oral morphology with prosthetic devices [e.g., tongue contact assisted floor]), which are important for the prevention of low nutrition and muscle weakness. Dysphagia precedes the onset of obvious disability. This reflects the patient's perception, but there was no significant difference between the two PNI groups, suggesting that few patients eligible for surgery have a significant functional decline.

Several limitations exist in this study. The present study was a single-center study with a retrospective design and a small sample size. As this study was conducted as exploratory research to formulate a hypothesis and contained a small sample size, the relationship between the PNI and tongue pressure could not be clearly determined. Selection bias exists because this is an observational study of surgery patients. In contrast, regarding the exclusion criteria for this study, we believe that the patients for whom surgery was discontinued were duplicate cancer cases with oral cancer, and that selection bias due to exclusion was less likely. Furthermore, nutritional status is influenced by several factors, including oral findings, smoking history, psychiatric and psychological status, socioeconomic status, daily activity performance, education level, processed food intake prevalence, and cultural differences in diet among countries. All of these factors were not investigated in this study. In addition, as the present study was a single-center, case-control retrospective study, it was not feasible to establish a causal relationship from the findings obtained. PNI is formulated based on ALB levels and lymphocyte counts. Therefore, caution should be exercised when interpreting patients with preoperative chemotherapy or severe inflammatory or autoimmune disease results. In epidemiological studies, dichotomous categories are used less frequently, and continuous variables or multiple categories are preferred [44,45]. Using multiple categories and creating ordinal variables is preferable to dichotomization. A decline in oral function results from tooth loss and reduced muscle activity. Much of the tooth loss is attributed to periodontal disease. Early detection and prevention of periodontal disease are important and require the discovery of biomarkers specific to periodontal disease. Known biomarkers for periodontal disease include CRP, IL-1β, TGF-β, VEGF, proteolytic degradation products (CTx), and matrix metalloproteinases (MMPs) [6–8]. These biomarkers are useful in the diagnosis and treatment of periodontal disease and can assess the progression of periodontal disease and the efficacy of treatment. However, these biomarkers should be easily measured. PNI, examined in this study, is one of the biomarkers and is easy to measure. On the other hand, there are limitations in diagnosing PNI alone, and it is necessary to make a comprehensive judgment based on symptoms and test results. Therefore, a large prospective validation study is warranted to confirm these results.

When the diagnostic criteria for OHF are considered a screening stage before functional impairment and malnutrition occur, it is possible to detect and respond early to the transition from OHF to malnutrition and to the flail cycle in the dental field. This early response can only be achieved through cooperation among multiple physicians and through cooperation among nurses, pharmacists, nutritionists, dental hygienists, and linguists, which will increase the contribution of dentistry to cooperation with multidisciplinary physicians and multidisciplinary professionals, such as nurses, pharmacists, nutritionists, dental hygienists, speech therapists, and physical therapists.

## Acknowledgments

We would like to thank Editage (www.editage.com) for English language editing.

## Author Contributions

**Conceptualization:** Atsushi Abe.

**Data curation:** Hiroki Hayashi.

**Formal analysis:** Yuya Otsuka, Kanae Shibata, Moeko Momokita, Shinichi Taniguchi.

**Investigation:** Moeko Momokita.

**Methodology:** Atsushi Abe.

**Project administration:** Atsushi Nakayama.

**Resources:** Kanae Shibata, Hiroki Hayashi, Shinichi Taniguchi.

**Software:** Yu Ito.

**Supervision:** Atsushi Abe.

**Validation:** Yoshihito Matsui.

**Visualization:** Yu Ito.

**Writing – original draft:** Atsushi Abe.

**Writing – review & editing:** Atsushi Nakayama.

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
