## [Decision Letter · Decision Letter 0]

10 Apr 2023

PONE-D-23-07423Relationship of preoperative oral hypofunction with prognostic nutritional index in gastric cancer: a case-control retrospective studyPLOS ONE

Dear Dr. Nakayama,

Thank you for submitting your manuscript to PLOS ONE. After careful consideration, we feel that it has merit but does not fully meet PLOS ONE’s publication criteria as it currently stands. Therefore, we invite you to submit a revised version of the manuscript that addresses the points raised during the review process.

We look forward to receiving your revised manuscript.

Kind regards,

Artak Heboyan, Ph.D.

Academic Editor

PLOS ONE

Journal Requirements:

Additional Editor Comments:

Dear authors,

The manuscript needs revision before processing further.

I noticed that reviewers suggested some references. Please, be informed that you can cite only very appropriate references and not accept those suggestions which are not very relevant. The editorial decision will be purely dependent on the quality of your paper and it cannot affect our final decision if you decide not to cite the recommended papers.

Best Regards

Reviewers' comments:

Reviewer's Responses to Questions

**Comments to the Author**

1. Is the manuscript technically sound, and do the data support the conclusions?

Reviewer #1: Yes

Reviewer #2: Partly

2. Has the statistical analysis been performed appropriately and rigorously? 

Reviewer #1: Yes

Reviewer #2: Yes

3. Have the authors made all data underlying the findings in their manuscript fully available?

Reviewer #1: Yes

Reviewer #2: Yes

4. Is the manuscript presented in an intelligible fashion and written in standard English?

Reviewer #1: Yes

Reviewer #2: Yes

5. Review Comments to the Author

Reviewer #1: Dear authors,

I revised the article “Relationship of preoperative oral hypofunction with prognostic nutritional index in gastric cancer: a case-control retrospective study”. The goal of this research was to investigate the relationship between the changes in preoperative oral function and the PNI, an immunonutritional index, in patients with gastric cancer, since a decline in preoperative oral function may contribute to postoperative complications. Furthermore, we investigated the correlation between PNI and the seven factors related to oral hypofunction.

The study is interesting and relevant, which enrolled 112 patients diagnosed with primary gastric cancer.

I endorse the publication. The study is straightforward and correct.

Reviewer #2: In the manuscript entitled: "Relationship of preoperative oral hypofunction with prognostic nutritional index in gastric cancer: a case-control retrospective study" the authors recruited patients with perioperative gastric cancer and examined preoperative oral hypofunction and its relationship with prognostic nutritional index. This case-control study analyzed 95 patients who underwent oral function management.

The authors found that significant differences in C-reactive protein, neutrophils, and tongue pressure (p<0.01) between the two groups. However, oral hygiene, oral dryness, occlusal force, tongue–lip motor function, masticatory function, and swallowing function were not significantly different. Multivariate analysis showed that C-reactive protein (p<0.01) and tongue pressure (p<0.05) were independent risk factors for oral hypofunction.

The authors concluded that oral function decreased in perioperative patients with gastric cancer, and decreased tongue pressure is associated with decreased prognostic nutritional index.

Major comments:

In general, the idea and innovation of this study regards the analysis of periodontitis and oral hypofunctions is interesting and novel because because the role these aspects in medicine are validated but further studies on this topic could be an innovative issue in this field could be open a creative matter of debate in literature by adding new information. Moreover, there are few reports in the literature that studied this interesting topic with this kind of study design.

The study was well conducted by the authors; However, there are some concerns to revise that are described below.

The introduction section resumes the existing knowledge regarding the important factor linked with periodontitis and related biomarkers.

However, as the importance of the topic, the reviewer strongly recommends, before a further re-evaluation of the manuscript, to update the literature through read, discuss and must cites in the references with great attention all of those recent interesting articles, that helps the authors to better introduce and discuss the role of some early biomarkers of periodontitis (TGF beta 1, circulating cells) which could better early diagnose the risk of both periodontitis and implant restoration especially in patients with periodontitis. 1) doi: 10.3892/ijmm.2012.1024. PMID: 22692760 2) doi: 10.2174/1874210601711010460. PMID: 28979575 3) doi: 10.3390/genes10121022. PMID: 31817862

The authors should be better specified, at the end of the introduction section, the rationale of the study and the aim of the study. In the central section, should better clarify inclusions and exclusions criteria of the selected sample.

The discussion section appears well organized with the relevant paper that support the conclusions, even if the authors should better discuss the relationship regarding the role exerted by relevant biomarkers of periodontitis in people who had gastric cancer and request also implant restoration. The conclusion should reinforce in light of the discussions.

In conclusion, I am sure that the authors are fine clinicians who achieve very nice results with their adopted protocol. However, this study, in my view does not in its current form satisfy a very high scientific requirement for publication in this journal and requests a revision before a futher re-evaluation of the manuscript.

Minor Comments:

Abstract:

- Better formulate the abstract section by better describing the aim of the study

Introduction:

- Please refer to major comments

Discussion

- Please add a specific sentence that clarifies the results obtained in the first part of the discussion

6. PLOS authors have the option to publish the peer review history of their article (what does this mean?). If published, this will include your full peer review and any attached files.

Reviewer #1: No

Reviewer #2: No

---

## [Author Response · Author response to Decision Letter 0]

12 May 2023

Reviewer(s)' Comments to Author:

Reviewer #1: 

Dear authors,

I revised the article “Relationship of preoperative oral hypofunction with prognostic nutritional index in gastric cancer: a case-control retrospective study”. The goal of this research was to investigate the relationship between the changes in preoperative oral function and the PNI, an immunonutritional index, in patients with gastric cancer, since a decline in preoperative oral function may contribute to postoperative complications. Furthermore, we investigated the correlation between PNI and the seven factors related to oral hypofunction.

The study is interesting and relevant, which enrolled 112 patients diagnosed with primary gastric cancer.

I endorse the publication. The study is straightforward and correct.

Reviewer #2: 

In the manuscript entitled: "Relationship of preoperative oral hypofunction with prognostic nutritional index in gastric cancer: a case-control retrospective study" the authors recruited patients with perioperative gastric cancer and examined preoperative oral hypofunction and its relationship with prognostic nutritional index. This case-control study analyzed 95 patients who underwent oral function management.

The authors found that significant differences in C-reactive protein, neutrophils, and tongue pressure (p<0.01) between the two groups. However, oral hygiene, oral dryness, occlusal force, tongue–lip motor function, masticatory function, and swallowing function were not significantly different. Multivariate analysis showed that C-reactive protein (p<0.01) and tongue pressure (p<0.05) were independent risk factors for oral hypofunction.

The authors concluded that oral function decreased in perioperative patients with gastric cancer, and decreased tongue pressure is associated with decreased prognostic nutritional index.

Major comments:

In general, the idea and innovation of this study regards the analysis of periodontitis and oral hypofunctions is interesting and novel because the role these aspects in medicine are validated but further studies on this topic could be an innovative issue in this field could be open a creative matter of debate in literature by adding new information. Moreover, there are few reports in the literature that studied this interesting topic with this kind of study design.

The study was well conducted by the authors; However, there are some concerns to revise that are described below.

The introduction section resumes the existing knowledge regarding the important factor linked with periodontitis and related biomarkers.

However, as the importance of the topic, the reviewer strongly recommends, before a further re-evaluation of the manuscript, to update the literature through read, discuss and must cites in the references with great attention all of those recent interesting articles, that helps the authors to better introduce and discuss the role of some early biomarkers of periodontitis (TGF beta 1, circulating cells) which could better early diagnose the risk of both periodontitis and implant restoration especially in patients with periodontitis. 

Response: We have cited three references to support the explanation on page 4, lines 47-51.

The authors should be better specified, at the end of the introduction section, the rationale of the study and the aim of the study. 

Response: Thank you for the valuable comment. We have modified the manuscript accordingly (page 5, lines 73-74)

In the central section, should better clarify inclusions and exclusions criteria of the selected sample.

Response: We apologize for the unclar inclusion and exclusions criteria. We have revised the manuscript accordingly (page 6, lines 80-86).

The discussion section appears well organized with the relevant paper that support the conclusions, even if the authors should better discuss the relationship regarding the role exerted by relevant biomarkers of periodontitis in people who had gastric cancer and request also implant restoration. 

Response: Thank you for pointing out this comment. We have added an explanation to pages 22-23, Lines 311-321.

The conclusion should reinforce in light of the discussions.In conclusion, I am sure that the authors are fine clinicians who achieve very nice results with their adopted protocol. However, this study, in my view does not in its current form satisfy a very high scientific requirement for publication in this journal and requests a revision before a futher re-evaluation of the manuscript.

Minor Comments:

Abstract:

- Better formulate the abstract section by better describing the aim of the study

Response: Thank you for the valuable comment, we have revised the manuscript accordingly. (Page 3, lines 24-26)

Introduction:

- Please refer to major comments

Discussion

- Please add a specific sentence that clarifies the results obtained in the first part of the discussion

Response: We realize that our original explanation was unclear. We have added the the meaning of the results and their implications. (Page 18, 222-223)

---

## [Decision Letter · Decision Letter 1]

18 May 2023

Relationship of preoperative oral hypofunction with prognostic nutritional index in gastric cancer: a case-control retrospective study

PONE-D-23-07423R1

Dear Dr. Nakayama,

We’re pleased to inform you that your manuscript has been judged scientifically suitable for publication and will be formally accepted for publication once it meets all outstanding technical requirements.

Kind regards,

Artak Heboyan, Ph.D.

Academic Editor

PLOS ONE

Additional Editor Comments (optional):

Reviewers' comments:

Reviewer's Responses to Questions

**Comments to the Author**

1. If the authors have adequately addressed your comments raised in a previous round of review and you feel that this manuscript is now acceptable for publication, you may indicate that here to bypass the “Comments to the Author” section, enter your conflict of interest statement in the “Confidential to Editor” section, and submit your "Accept" recommendation.

Reviewer #2: All comments have been addressed

2. Is the manuscript technically sound, and do the data support the conclusions?

Reviewer #2: Yes

3. Has the statistical analysis been performed appropriately and rigorously? 

Reviewer #2: Yes

4. Have the authors made all data underlying the findings in their manuscript fully available?

Reviewer #2: Yes

5. Is the manuscript presented in an intelligible fashion and written in standard English?

Reviewer #2: Yes

6. Review Comments to the Author

Reviewer #2: The authors have well addressed to all of the issues raised by the reviewer. No further adjustment are needed.

7. PLOS authors have the option to publish the peer review history of their article (what does this mean?). If published, this will include your full peer review and any attached files.

Reviewer #2: No

---

## [Editor Report · Acceptance letter]

23 May 2023

PONE-D-23-07423R1 

Relationship of preoperative oral hypofunction with prognostic nutritional index in gastric cancer: a case-control retrospective study 

Dear Dr. Nakayama:

I'm pleased to inform you that your manuscript has been deemed suitable for publication in PLOS ONE. Congratulations! Your manuscript is now with our production department. 

Kind regards, 

on behalf of

Dr. Artak Heboyan 

Academic Editor

PLOS ONE